# Self-assembly coupled to liquid-liquid phase separation

**Michael F. Hagan** *, **Farzaneh Mohajerani**

Martin A. Fisher School of Physics, Brandeis University, Waltham, Massachusetts, United States of America

* hagan@brandeis.edu

## Abstract

Liquid condensate droplets with distinct compositions of proteins and nucleic acids are widespread in biological cells. While it is known that such droplets, or compartments, can regulate irreversible protein aggregation, their effect on reversible self-assembly remains largely unexplored. In this article, we use kinetic theory and solution thermodynamics to investigate the effect of liquid-liquid phase separation on the reversible self-assembly of structures with well-defined sizes and architectures. We find that, when assembling subunits preferentially partition into liquid compartments, robustness against kinetic traps and maximum achievable assembly rates can be significantly increased. In particular, both the range of solution conditions leading to productive assembly and the corresponding assembly rates can increase by orders of magnitude. We analyze the rate equation predictions using simple scaling estimates to identify effects of liquid-liquid phase separation as a function of relevant control parameters. These results may elucidate self-assembly processes that underlie normal cellular functions or pathogenesis, and suggest strategies for designing efficient bottom-up assembly for nanomaterials applications.

**Data Availability Statement:** The Matlab scripts used to generate the figures, the associated data files, and the figure files have been deposited in the Open Science Framework database, and are available at https://osf.io/mr9a3/.

## Author summary

Liquid-liquid phase separation describes the de-mixing of a fluid into 'compartments' with different compositions, such as the separation of oil and water. Liquid-liquid phase separation occurs within biological cells, allowing different chemical reactions to occur within different compartments. One such reaction is self-assembly, in which proteins and other biomolecules organize into larger, more complex structures, such as a virus particle. It has recently been shown that many viruses self-assemble in liquid-liquid phase-separated compartments within their host cells. However, the effects of liquid-liquid phase separation on self-assembly, and how it may facilitate the formation of virus particles or other biological complexes, are not understood. We develop theoretical models, which show that liquid-liquid phase separation can make self-assembly occur significantly faster, and make it more likely to result in properly assembled particles. The models also reveal the mechanisms underlying these effects, showing that by locally concentrating subunits, phase separation can accelerate assembly while simultaneously preventing the system from running out of subunits before assembly completes. These findings could enable

**Funding:** This work was supported by the National Institutes of Health (RO1GM108021 to MFH and FM) and the National Science Foundation (DMR-2011846 to FM). The funders had no role in study design, data collection and analysis, decision to publish, or preparation of the manuscript.

**Competing interests:** The authors have declared that no competing interests exist.

new strategies to prevent or treat viral infections. More broadly, these insights can be applied to understand other self-assembly reactions in biological cells.

## Introduction

The self-assembly of basic subunits into larger structures with well-defined architectures underlies essential functions in biological organisms, where examples of assembled structures include multi-protein filaments such as microtubules or actin [1, 2], scaffolds for vesicular budding [3–9], the outer shells or 'capsids' of viruses [10–16], and bacterial microcompartments [17–22] or other proteinaceous organelles [23–27]. However, achieving efficient and high fidelity assembly into target architectures requires precisely tuned subunit interaction strengths and concentrations due to competing thermodynamic and kinetic effects (e.g. [28–47]). The need for such precision could severely constrain the use of assembly for biological function or human engineered applications. Biological organisms employ multiple modes of biochemical and physical regulation to overcome this limitation. In this article, we investigate one such mode—how spatial heterogeneity due to formation of biomolecular condensates, or liquid-liquid phase-separation (LLPS), can dramatically enhance the speed and robustness of self-assembly.

While membranous organelles play a prominent role in compartmentalizing eukaryotic cells, it is now clear that condensates act as 'membrane-less compartments' to spatially organize cellular interiors within all kingdoms of life (e.g. [48–69]). These compartments are implicated in diverse cellular functions, including transcriptional regulation [53, 70–73], formation of neuronal synapses [74–76], enrichment of specific proteins and nucleic acids [77–82], cellular stress responses [83–86], and cell division [79, 87]. In addition to the roles of condensates in normal cellular function, pathogenic viruses generate or exploit LLPS during various stages of their life cycles [88–93]. Most relevant to this article, many viruses undergo assembly and/or genome packaging within phase-separated compartments known as virus factories, replication sites, Negri bodies, inclusion bodies, or viroplasms [88–108]. *In vitro* studies show that viral nucleocapsid proteins and RNA molecules undergo LLPS (e.g. [99, 106, 109–111]), and that LLPS accelerates assembly of nucleocapsid-like particles [99]. It is hypothesized that viruses exploit LLPS to avoid host immune responses and coordinate events such as RNA replication, capsid protein translation, assembly, and genome packaging. However, the mechanisms underlying these events are poorly understood.

In addition to viruses, other examples of biological self-assembly coupled to LLPS include the formation of clathrin cages to mediate endocytosis [112]; post-synaptic densities [113] and pre-synaptic vesicles release sites (active zones) [114, 115] at neuronal synapses; observations that condensates can both accelerate and suppress aggregation of $\alpha$-synuclein [116], and actin assembly in polypeptide coacervates [117].

Multiple lines of evidence suggest that condensate formation is driven by favorable interactions among their constituents combined with unfavorable interactions with the bulk exterior cytoplasm or nucleoplasm. Although condensation may be driven, destabilized, or regulated by diverse nonequilibrium effects (e.g. [54, 70–73, 118–121]), equilibrium thermodynamics provides a starting point to model their stability, and their formation is frequently described as LLPS [48, 49, 52, 52–60, 69, 74, 85, 118–131]. Henceforth, we will use the term LLPS, keeping in mind that nonequilibrium effects may also be present. Consistent with equilibrium phase coexistence, the composition of the compartment interior can significantly differ from that of the cytoplasm. Thus, LLPS can provide significant spatiotemporal control over reaction

processes by concentrating and colocalizing specific sets of subunit species that preferentially partition into the compartment.

These capabilities potentially enable LLPS to strongly regulate self-assembly. Yet, despite recent intense investigations into LLPS, its coupling to assembly has yet to be fully explored. Previous simulations showed that the condensed enzyme complex that forms the interior cargo of bacterial microcompartments can promote nucleation and control the size of the exterior protein shell [43, 45, 132–134]. Most closely related to our work, Refs. [116, 135–137] recently showed that the presence of a compartment can significantly accelerate irreversible protein aggregation into linear fibrils.

Here, we investigate the effects of LLPS on *reversible* self-limited assembly into target structures with finite sizes and well-defined architectures. Self-limited assembly from bulk solution is constrained by competing thermodynamic and kinetic effects—subunit interactions must be sufficiently strong and geometrically precise to stabilize the target structure, but overly high interaction strengths or subunit concentrations lead to kinetic traps (e.g. [28–47]). Avoiding such kinetic traps imposes a 'speed limit' on assembly from bulk solution [41, 42, 138].

Using a master equation description of assembly, we show that these thermodynamic and kinetic constraints can be simultaneously satisfied by spatial heterogeneity due to phase-separated compartments. We find that LLPS can significantly accelerate assembly nucleation, consistent with previous studies of irreversible assembly [116, 135–137], but also induces kinetic traps that *slow* assembly in certain parameter regimes. Crucially though, by enhancing nucleation only within spatially localized regions, LLPS significantly expands the range of subunit concentrations and interaction strengths over which such kinetic traps are avoided, thus promoting assembly robustness. This effect can increase by orders of magnitude the maximum rate of *productive* assembly into the ordered target structure. The extent of assembly acceleration and robustness enhancement are nonmonotonic functions of the key control parameters: the compartment size, and the partition coefficient of subunits between the compartment and the bulk cytoplasm. We present simple scaling estimates that capture the effect of LLPS on assembly, and reveal the underlying mechanisms that enable regulation. For example, the bulk solution acts as a "buffer" that steadily supplies free subunits to the compartment to enable rapid assembly without kinetic traps. Although we particularly focus on self-limited assembly processes that lead to finite-sized structures, our models are general and many results also apply to unlimited assembly or crystallization.

## Methods

### Model

We have developed a minimal model to describe assembly in the presence of one or more liquid droplets coexisting with a background solution of different composition (Fig 1). We are motivated by processes such as virus assembly, in which viral proteins, nucleic acids and other viral components, and possibly some host proteins phase separate to form liquid compartments within the cellular cytoplasm. For this initial study, we consider only one assembling species in the limit that the assembly subunits comprise a small fraction of the compartment mass, and thus the size and composition of the compartment can be treated as independent of the subunit concentration.

We consider a system of subunits that form self-limited assemblies with optimal size $N$. The subunits are immersed in a multicomponent solvent which is in a state of phase coexistence, with stoichiometry such that there are one or more small compartments rich in one (or more) solvent species coexisting with a much larger background rich in the other species. While we consider protein subunits undergoing assembly, the solvent could be comprised of proteins,

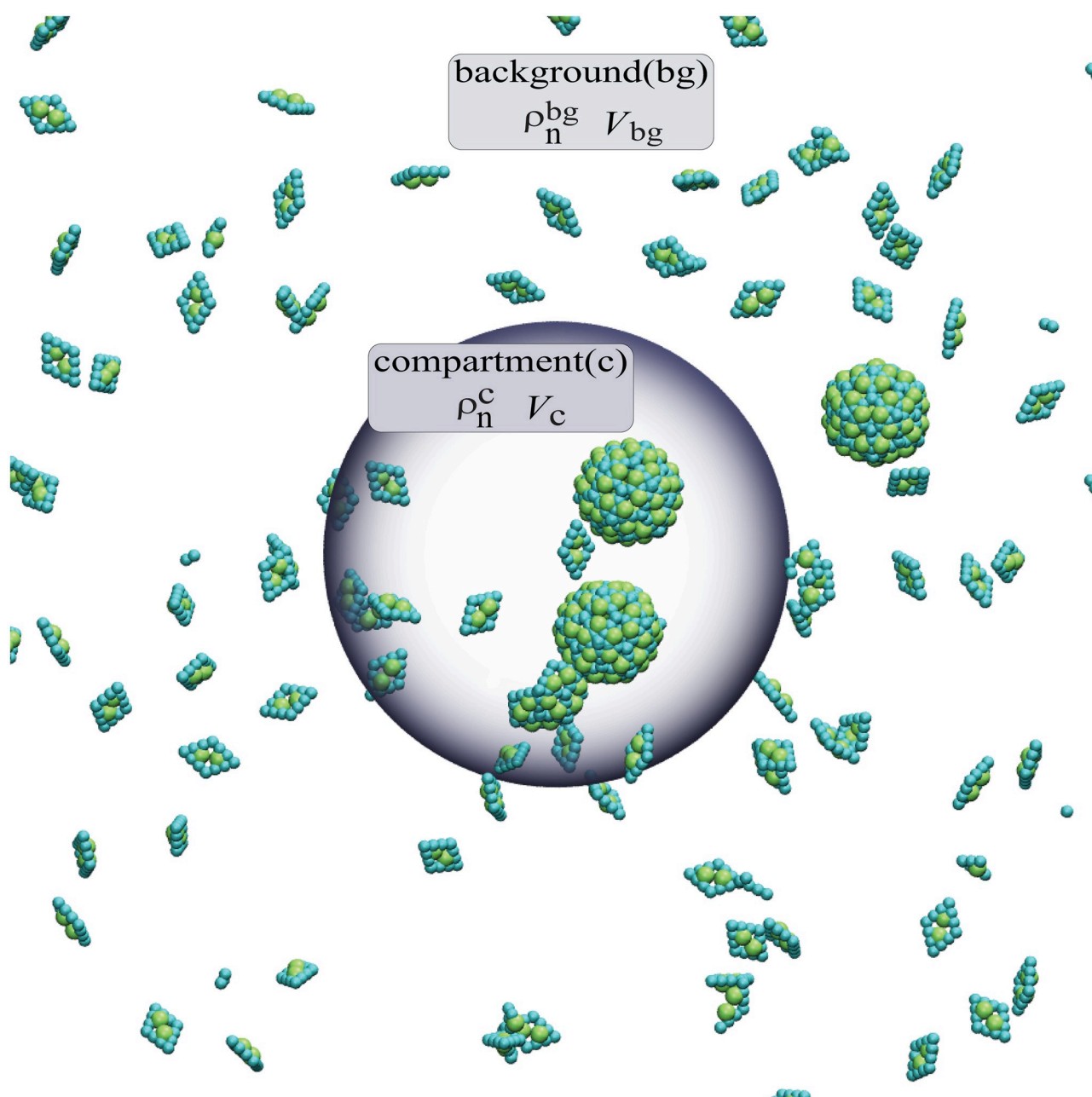

**Fig 1. Schematic of the model.** Subunits exchange between bulk and the phase-separated compartment (gray sphere), with equilibrium concentrations related by $K_c = \rho_1^c / \rho_1^{bg}$. Assembly can occur anywhere in the system, but occurs preferentially in the compartment when $K_c > 1$ due to the enhanced local subunit concentration. The volumes of the bulk $V_{bg}$ and compartment $V_c$ are related by $V_r = V_c / V_{bg}$.

nucleic acids, or other macromolecules. The distinction is that 'subunits' form ordered (para) crystalline structures such as a capsid, while the primary compartment constituents remain amorphous and (for the regime we consider) in the liquid phase.

Our model can be viewed as a minimal starting point motivated by biological assembly coupled to LLPS. For example, in the model proposed for rotavirus assembly, two nonstructural rotavirus proteins (NSP2 and NSP5) undergo LLPS to form a viroplasm. The capsid proteins (VP2 and VP6) partition favorably into the viroplasm due to weak multivalent interactions

with NSP2 and NSP5 [94, 98, 104, 107]. Here, NSP2 and NSP5 would correspond to the primary compartment constituents, and the capsid proteins (VP2 and VP6) correspond to the assembly subunits. We note that the rotavirus genomic RNA is also driven to partition into the viroplasm through interactions between RNA binding proteins and NSP2/5, but we do not explicitly consider assembly around RNA in this work to simplify the model. Our assumption of low concentration of assembly subunits in the compartment noted above is consistent with concentrations of VP2/6 in the viroplasm that are small compared to those of NSP2/5 [94, 98, 104, 107].

The driving force for subunits to enter the compartment phase is characterized by the partition coefficient $K_c$, which at equilibrium satisfies

$$K_c = \rho_1^c / \rho_1^{bg} \qquad (1)$$

with $\rho_1^c$ and $\rho_1^{bg}$ the subunit concentrations in the compartment and background. The partition coefficient is related to the change in solvation free energy $g_c$ for a subunit that transitions from the background to the compartment as $K_c = e^{-\beta g_c}$ with $\beta = 1/k_B T$ with $k_B T$ the thermal energy. Applying standard dilute solution thermodynamics will result in a subunit solvation free energy difference with the form (see Weber et al. [120]) $g_c \propto n_s(\Delta\chi)(\Delta\phi)$, where $\Delta\phi$ is the difference in solvent composition between the background and compartment, $\Delta\chi$ is the difference in interaction strength (parameterized by the Flory $\chi$ parameter) for an interaction site on the subunit between the background and compartment, and $n_s$ is the number of interaction sites per subunit. The key point is that even for relatively weak interactions, a subunit with multivalency of $n_s \gtrsim 10$ could have a partition coefficient as large as $K_c \sim 10^4 - 10^5$, although $K_c \sim 100$ may be a typical value [54].

We denote the volumes of the compartment and background as $V_c$ and $V_{bg}$, which are related to the total system volume by $V_{tot} = V_c + V_{bg}$. We will present results in terms of the compartment size ratio, $V_r \equiv V_c/V_{bg}$. In most biological systems or *in vitro* experiments, the compartment volume will be small compared to the background, $V_r \ll 1$. For this work we assume a fixed total subunit concentration $\rho_T$. For simplicity we will typically consider a single compartment, but we also discuss the case of multiple compartments, which might arise due to microphase separation or arrested phase separation.

## Typical and minimal compartment sizes

Let us consider a single compartment in a eukaryotic cell with radius $R_{cell} = 10$ μm. At our default compartment volume ratio of $V_r = 10^{-3}$, the compartment radius is $R_c = 1$ μm. Assuming a typical protein subunit with mass 30 kDa and a volume of about 50 nm³, requiring a volume fraction of subunits ≤0.01 results in the total number of proteins in the compartment $N_c \lesssim 10^6$, which is large compared to a typical assembly size of 100–1000 subunits, and sufficiently large that finite number fluctuations can be neglected, at least to a first approximation. Along these lines, defining a 'minimum' compartment size as the smallest compartment containing $N_{min} \sim 1000$ subunits gives $R_{min} \approx 0.1$ μM. Since compartment radii scale with subunit number $\propto N^{(1/3)}$, these estimates are insensitive to the assembly size.

## Master equation models for capsid assembly kinetics

To simulate the assembly kinetics, we adapt the rate equation description originally developed by Zlotnick and coworkers [28, 29, 139] and used by others [138, 140] to describe the self-assembly of 2D polymers (capsids) in bulk solution. Denoting the concentration of an intermediate with $n$ subunits in either phase as $\rho_n^\alpha$ with $\alpha$ = c, bg, the time evolution of intermediate

concentrations is given by:

$$
\begin{aligned}
\frac{d\rho_1^\alpha}{dt} = & \; -2f_1\left(\rho_1^\alpha\right)^2 + b_2\rho_2^\alpha \\
& + \left(\sum_{n=2}^{N-1} -f_n\rho_n^\alpha\rho_1^\alpha + b_n\rho_n^\alpha\right) + b_N\rho_N^\alpha + \mathcal{D}_1^\alpha \\
\frac{d\rho_n^\alpha}{dt} = & \; f_{n-1}\rho_1^\alpha\rho_{n-1}^\alpha - (f_n\rho_1^\alpha + b_n)\rho_n^\alpha \\
& + b_{n+1}\rho_{n+1}^\alpha + \mathcal{D}_n^\alpha \quad \text{for } n = 2\ldots N-1 \\
\frac{d\rho_N^\alpha}{dt} = & \; f_{N-1}\rho_1^\alpha\rho_{N-1}^\alpha - b_N\rho_N^\alpha + \mathcal{D}_N^\alpha
\end{aligned}
\tag{2}
$$

with the diffusive exchange between the phases given by (see Section A in S1 Text and Refs. [135, 137])

$$
\begin{aligned}
\mathcal{D}_n^c = & \; \frac{1}{V_c}k_{\mathrm{DL}}(n)\left(\rho_n^{\mathrm{bg}} - \rho_n^c/K_c^n\right) \\
\mathcal{D}_n^{\mathrm{bg}} = & \; -V_r\mathcal{D}_n^c
\end{aligned}
\tag{3}
$$

and $f_n$ and $b_n$ as the association and dissociation rate constants for intermediates of size $n$. We set the initial condition as $\rho_1^{\mathrm{bg}}(0) = K_{\mathrm{eff}}\rho_{\mathrm{T}}$, $\rho_1^c(0) = K_cK_{\mathrm{eff}}\rho_{\mathrm{T}}$, and $\rho_n^c(0) = \rho_n^{\mathrm{bg}}(0) = 0 \quad \forall n > 1$.

We have made several assumptions to simplify the models, based on previous work (e.g [28, 29, 39, 41, 138–140]). First, we assume that there is only one 'average' intermediate structure for each size $n$. Second, we assume that only individual subunits can associate to or dissociate from an intermediate. This assumption is based on the fact that particle-based computer simulations show that, at the dilute conditions typical of productive assembly reactions, most assembly events involve association of individual subunits [138], and that extending Eq (2) to allow for binding of higher-order oligomers does not qualitatively change the results (see the supplemental material of Ref. [138]). Third, we assume that the domain composition is independent of subunit concentration and assembly. Similarly, we assume that the diffusion coefficient is independent of subunit concentration. These simplifications are based on the assumption of low concentrations of subunits in the compartment and that the other compartment constituents are macromolecules that typically have equal or larger molecular weights as the subunits. To focus on effects of competing reactions on assembly, we also neglect the possible dependence of diffusion coefficients on intermediate size or $g_c$, considered in Refs. [136, 137] respectively. The model can be readily extended to account for these effects.

The most important simplification is that we neglect the possibility of malformed (off-pathway) structures. While this is a good assumption under productive assembly conditions, particularly when subunit-subunit interactions have high orientational specificity, malformed structures can lead to kinetic traps at high concentrations or binding affinities [32–34, 36–42, 141–143]. This effect will be considered by performing particle-based simulations in a future work, and we discuss its implications in the Conclusions section.

To complete the Master equation description we must specify the transition rates between intermediates. We consider two models, which consider different dependencies of rates on the partial capsid size [138].

**Nucleation and growth model (NG).**   We start with a simple generic model for a nucleated self-assembly process denoted as the 'nucleation and growth (NG) model' [138]. This can describe linear assembly with nucleation (e.g. assembly of a helical viral capsid or the

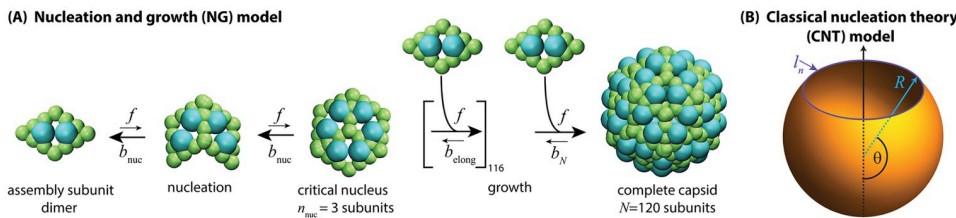

**(A) Nucleation and growth (NG) model**

assembly subunit dimer · nucleation · critical nucleus $n_{\text{nuc}}$ = 3 subunits · growth · complete capsid $N$=120 subunits

**(B) Classical nucleation theory (CNT) model**

**Fig 2. Schematics of the two assembly pathway models. (A) Nucleation and growth (NG) assembly pathway**, Eq (4). There is one average intermediate for each size $n$. Subunits (capsid protein dimers in the schematic) associate or dissociate to intermediates with association rate $f$ and dissociation rates: $b_{\text{nuc}} > f\rho_1$ below the critical nucleus size ($n_{\text{nuc}}$, a trimer of dimers in the schematic), $b_{\text{elong}} < f\rho_1$ during elongation, and $b_N < b_{\text{elong}}$ for dissociation of a subunit from a complete capsid. The rates $f$ and $b_i$ with $i \in \{\text{nuc, elong}, N\}$ are related to the subunit binding free energy $g_i$ by detailed balance as $v_0 b_i = f \exp(\beta g_i)$ with $v_0$ the standard state volume and $|g_{\text{nuc}}| < |g_{\text{elong}}| < |g_N|$ accounting for the increase in number of contacts per subunit as the intermediate size grows. **(B) Classical nucleation theory (CNT) assembly pathway.** Capsid intermediates are represented as continuum elastic partial spherical shells, with the total binding free energy given by Eqs (5) and (6).

equilibrium assembly of an actin filament), or polyhedral shell assembly with an initial nucleation step, followed by assembly along a single growth front until the shell closes on itself [29, 138, 140] (see Fig 2A).

We consider a system of capsid protein subunits with total concentration $\rho_T$ that start assembling at the time $t = 0$. We assume that the rate constants are the same in the compartment and background, so we simplify the presentation by omitting the specification of phase in this subsection. Our reaction is given by:

$$1 \underset{b_{\text{nuc}}}{\overset{f\rho_1}{\rightleftharpoons}} 2 \underset{b_{\text{nuc}}}{\overset{f\rho_1}{\rightleftharpoons}} \dots \underset{b_{\text{nuc}}}{\overset{f\rho_1}{\rightleftharpoons}} n_{\text{nuc}-1} \underset{b_{\text{elong}}}{\overset{f\rho_1}{\rightleftharpoons}} n_{\text{nuc}} \dots \underset{b_N}{\overset{f\rho_1}{\rightleftharpoons}} N \tag{4}$$

where $b_i$ is the dissociation rate constant (with $i = \{\text{nuc, elong}, N\}$), which is related to the forward rate constant by detailed balance as $v_0 b_i = f \exp(\beta g_i)$, with $g_i$ the change in interaction free energy upon subunit association to a partial capsid and $v_0$ the standard state volume. The nucleation and elongation phases are distinguished by the fact that association in the nucleation phase has an unfavorable free energy change, $g_{\text{nuc}} - k_B T \ln(\rho_1 v_0) > 0$, while association in the elongation phase is favorable, $g_{\text{elong}} - k_B T \ln(\rho_1 v_0) < 0$. For the moment, we assume that there is a single critical nucleus size $n_{\text{nuc}}$.

For most results in this article, we will set $g_{\text{nuc}} = -4k_B T$ and $g_{\text{elong}} = -17k_B T$ and $g_N = 2g_{\text{elong}}$. The small value of $g_{\text{nuc}}$ relative to $g_{\text{elong}}$ accounts for the fact that the first few subunits to associate make fewer and/or less favorable contacts than subunits in larger intermediates, giving rise to a nucleation barrier, while the large value of $g_N$ accounts for the fact that in many assembly geometries the last subunit makes the largest number of contacts upon associating. We have chosen values of $g_{\text{nuc}}$ and $g_{\text{elong}}$ to be roughly consistent with binding affinity values estimated for virus capsid assembly [28, 30, 31, 144], but the results do not qualitatively change for other affinity values within a given assembly regime.

**Classical nucleation theory model (CNT).** To test whether our conclusions depend qualitatively on the model geometry, we also consider transition rates based on the 'classical nucleation theory (CNT)' model for icosahedral capsids suggested by Zandi et al. [145]. In this model, assembly intermediates are represented as partial spheres that are missing a spherical cap. With the parameterization shown in Fig 2B, the total capsid size is $N = 4\pi R^2/a_0$ with $R$ the capsid radius and $a_0$ the subunit area, and intermediate sizes are given by $n = N(1 - \cos \theta)/2$. Subunits along the perimeter of the missing cap have fewer interactions than those in the shell interior, leading to a line tension $\sigma$, and the total binding free energy for an intermidate with $n$

subunits is

$$G_n = ng_{\text{sub}} + \sigma l_n \tag{5}$$

with the perimeter of the missing spherical cap given by

$$l_n = l_0 2[\pi n(N - n)/N]^{1/2}. \tag{6}$$

with $g_{\text{sub}}$ the binding free energy per subunit in a complete capsid and $l_0$ the diameter of a subunit. Following previous work [138, 145], we set the line tension to $\sigma = -g_{\text{sub}}/2l_0$, so that a subunit adding to the perimeter of the capsid satisfies half of its contacts on average. We assume that the forward rate constant is proportional to the number of subunits on the perimeter, $f_n = f_0 l_n/l_0$, with $f_0$ the association rate constant for a single binding site, and we set $a_0 = v_0^{2/3}$ and $l_0 = v_0^{1/3}$.

The key difference between the CNT and NG models is the dependence of the critical nucleus size on solution conditions. For the NG model the critical nucleus size $n_{\text{nuc}}$ is constant, provided $\exp(g_{\text{nuc}}/k_{\text{B}}T) < \rho_1 v_0 < \exp(g_{\text{elong}}/k_{\text{B}}T)$. For the CNT model, the critical nucleus size varies with subunit concentration and interaction strengths, and is given by the maximum in $k_{\text{B}}T \log(\rho_1 v_0)n + G_n$, or [145]

$$n_{\text{nuc}} = 0.5N\left(1 - \frac{\Gamma}{(\Gamma^2 + 1)^{1/2}}\right) \tag{7}$$

with $\Gamma = [g_{\text{sub}} - \ln(\rho_1 v_0)]/\sigma l_0$. Thus, the critical nucleus size continually changes over time during an assembly process for the CNT model as subunits are depleted, whereas it is constant until the very late stages of a NG assembly process.

## Results and discussion

### Effects of LLPS on self-assembly equilibrium

We begin by calculating how the equilibrium yield of self-assembled structures depends on subunit concentrations and interaction strengths, as well as the two key control parameters for subunit partitioning into the compartment: the partition coefficient $K_{\text{c}}$ and the compartment size ratio $V_{\text{r}}$.

At equilibrium the subunit concentrations in the compartment and background are related to each other by $K_{\text{c}}$, and to the total subunit concentration $\rho_{\text{T}}$ by mass conservation, giving

$$\rho_1^c = K_{\text{eff}} K_{\text{c}} \rho_{\text{T}} \tag{8}$$

with

$$K_{\text{eff}} \equiv \frac{1 + V_{\text{r}}}{1 + K_{\text{c}} V_{\text{r}}}. \tag{9}$$

**Equilibrium assembly yield.** We now calculate the effect of the compartment on assembly yields, using the well-justified approximation that intermediates have very low concentrations at equilibrium for self-limited assembly [41, 146]. Thus we consider a two-state system, with finite concentrations of only free subunits and complete assemblies with $N$. Mass conservation then gives

$$(1 + V_{\text{r}})\rho_{\text{T}} = V_{\text{r}}(\rho_1^c + N\rho_N^c) + (\rho_1^{\text{bg}} + N\rho_N^{\text{bg}}) \tag{10}$$

where $\rho_N^c$ and $\rho_N^{bg}$ are the concentrations of assemblies in the compartment and background. At equilibrium these are related to the free subunit concentration by the law of mass action [41, 147]

$$\rho_N^\alpha = \ (\rho_1^\alpha)^N e^{-\beta N g_{sub}} \tag{11}$$

with $\alpha$ = dom, bg and $g_{sub}$ as the per-subunit interaction energy within a complete assembly (which we assume is the same in the compartment and background). Eqs (10) and (11) can be easily solved numerically. However, we can make the results more transparent by following previous analysis for homogeneous assembly [41, 148] and writing the fraction of subunits in assemblies as

$$\begin{aligned}
x_N^{bg} &\equiv \ \left(N\rho_N^{bg}V_{bg}\right)/(\rho_T V_{tot}) \\
x_N^c &\equiv \ \left(N\rho_N^c V_c\right)/(\rho_T V_{tot}) \approx \left(N\rho_N^c V_r\right)/\rho_T \\
\chi_N &= \ \chi_N^c + \chi_N^{bg}.
\end{aligned} \tag{12}$$

This simplifies to

$$\frac{(x_N)^{\frac{1}{N-1}}}{(1-(x_N))^{\frac{N}{N-1}}} = \left(N\frac{1+V_r K_c^N}{1+V_r}\right)^{\frac{1}{N-1}} \tag{13}$$
$$\times (K_{eff})^{\frac{N}{N-1}} e^{-\beta g_{sub}\frac{N}{N-1}}\rho_T$$

$$x_N^c = \ \frac{V_r K_c^N}{1+V_r K_c^N}x_N. \tag{14}$$

In the limit of large optimal assembly size $N \gg 1$, Eq (13) satisfies the following asymptotic limits:

$$x_N^c \approx \begin{cases} 1-\dfrac{\rho_{CAC}}{\rho_T} & \text{for } \rho_T \gg \rho_{CAC} \\[3mm] \left(\dfrac{\rho_T}{\rho_{CAC}}\right)^N & \text{for } \rho_T \ll \rho_{CAC} \end{cases} \tag{15}$$

with $\rho_{CAC}$ the *critical assembly concentration* (CAC) given by (assuming $K_c^N V_r \gg 1$, so that all assembly occurs in the compartment)

$$\begin{aligned}
\rho_{CAC} &\approx \ (K_c K_{eff})^{-1}\left(\frac{V_r}{V_r+1}\right)^{-1/N}\rho_{CAC}^0 \\
&\approx \ \rho_{CAC}^0/(K_c K_{eff})\text{ for } N \gg 1, V_r \ll 1
\end{aligned} \tag{16}$$

with

$$\rho_{CAC}^0 \ \equiv N^{-1/N}e^{\beta g_{sub}} \tag{17}$$

as the CAC in a system without coupling to LLPS (i.e. $V_r = 0$ or $K_c = 1$). In all subsequent expressions, we will write the limit of no LLPS or $K_c \to 1$ with a superscript '0'. The last expression in Eq (16) assumes $V_r \ll 1$ and shows that LLPS reduces the CAC by a factor $K_c K_{eff}$. Then

using Eq (8) we arrive at the simple result that significant assembly occurs when the *total* subunit concentration $\rho_T$ exceeds the *local* CAC within the compartment.

To obtain further insight, we note that $K_c K_{\text{eff}} \approx (V_r + 1/K_c)^{-1}$, giving the asymptotic limits

$$\rho_{CAC}^0 / \rho_{CAC} \approx \begin{cases} 1/V_r & \text{for } K_c V_r \ll 1 \\ K_c & \text{for } K_c V_r \gg 1 \end{cases}. \tag{18}$$

and that maximal enhancement of equilibrium assembly is achieved when $K_c \gtrsim V_r$.

**Selectivity and spatial control over assembly.** We can draw two important conclusions from Eq (16). First, the presence of a compartment allows assembly under conditions where there is no bulk assembly (Fig 3). Second, there is a range of total subunit concentrations $\propto K_c K_{\text{eff}}$ over which assembly occurs only in the compartment, thus allowing for spatial control over assembly. As a measure of the extent to which LLPS can spatially control assembly, we define *selectivity* as $x_{\text{selec}} \equiv \frac{V_c \rho_N^c}{V_c \rho_N^c + V_{bg} \rho_N^{bg}}$. The equilibrium selectivity is then given by

$$x_{\text{selec}}^{\text{equil}} = \frac{V_r K_c^N}{V_r K_c^N + 1}. \tag{19}$$

We thus see that even a very small partition coefficient leads to strong equilibrium selectivity due to the high-valence nature of an assembled capsid. In particular, an assembled capsid has $\sim N$ interactions with compartment components, but only has three translational degrees of freedom suppressed by partitioning into the compartment volume. However, if assembled capsids and large intermediates are not able to rapidly exchange between the compartment and background [137], the selectivity at finite times may be under kinetic control.

## Effect of LLPS on self-assembly kinetics

### Master equation results

**Assembly kinetics and yields.** Figs 3 and 4 show the effect of LLPS on assembly kinetics, as measured by the fraction of subunits in complete capsids ($x_N$), obtained by numerically integrating the Master equation (Eq 2) with the NG model (Eq 4). Fig 4A shows $x_N$ as a function of time for several initial subunit concentrations $\rho_T$ in the absence of LLPS. There is an initial lag phase during which intermediate populations build up to a quasi-steady-state, followed by rapid appearance of complete capsids, and then eventually saturation as free subunits are depleted. The duration of the lag phase decreases as $1/\rho_T$.

Importantly, the rate of capsid production is nonmonotonic with respect to $\rho_T$—yields of complete capsids are suppressed for the highest concentration shown ($\rho_T = 4M$) by the *monomer starvation* kinetic trap arising from depletion of free subunits before capsids finish assembling. These results are discussed further in section Assembly timescales without LLPS and Refs. [41, 138]. This kinetic trapping effect is responsible for the low values of $x_N$ at high concentrations for finite-time results in Fig 3.

Figs 4B and 3A show how the assembly kinetics is changed by LLPS. With the lowest concentration shown in Fig 4A ($\rho_T = 0.2\,\mu M$), $x_N$ is shown as a function of time for increasing values of the partition coefficient $K_c$. We see that the yields and assembly rates increase dramatically, with the duration of the lag phase decreasing and the maximum rate of capsid production (corresponding to the nucleation rate) increasing with $K_c$. To give a more comprehensive picture, Fig 3A shows $x_N$ as a function of both $K_c$ and $\rho_T$. We see that assembly occurs at lower concentrations as $K_c$ increases, and that LLPS increases the range of concentrations over which productive assembly occurs, particularly for $K_c$ of $\mathcal{O}(10)$. Similarly, Fig 3B and 3C

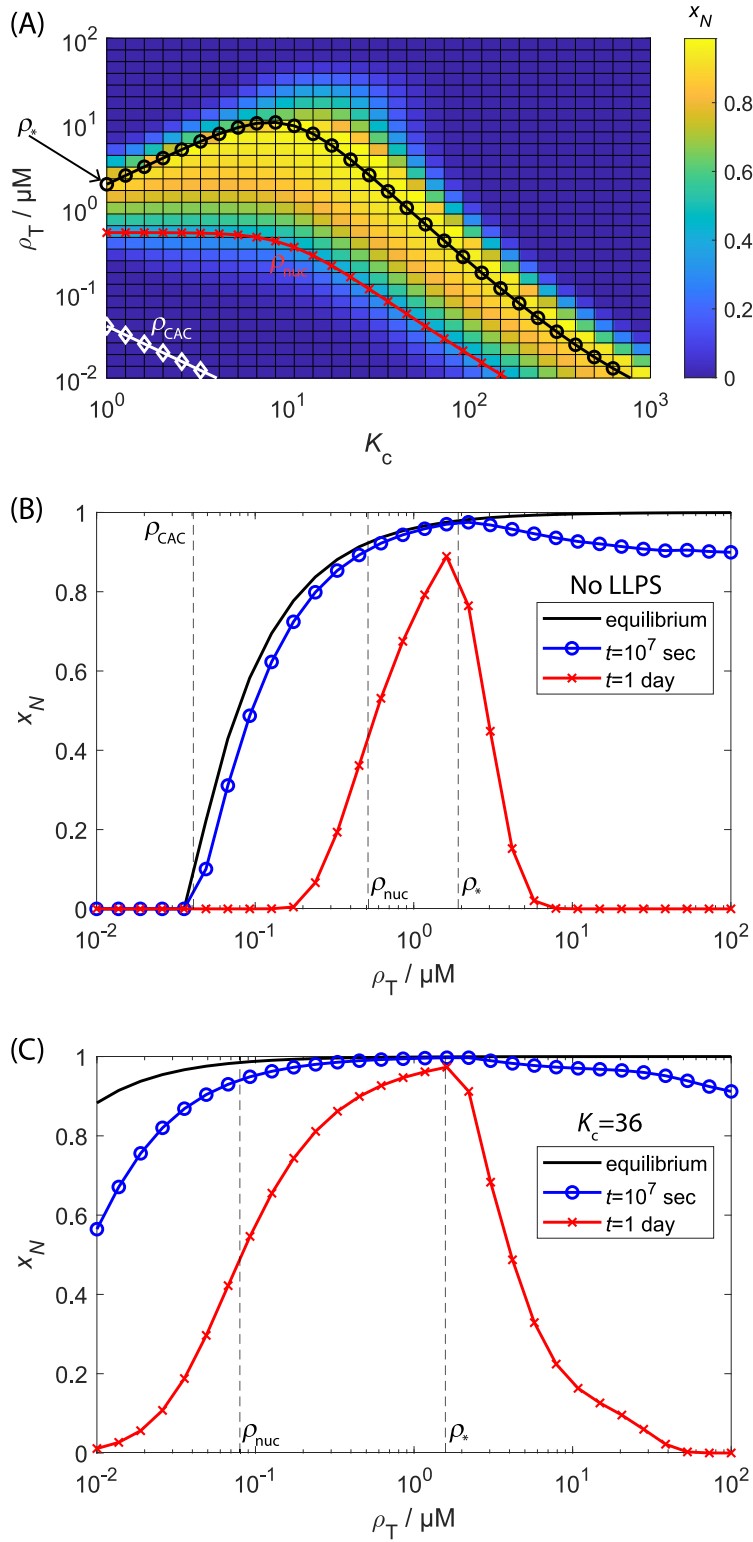

**Fig 3. Effect of LLPS on the equilibrium and finite-time yields of self-assembly. (A)** The heat map shows the mass fraction of subunits in capsids ($x_N$) as a function of the compartment partition coefficient ($K_c$) and total subunit concentration ($\rho_T$) computed from the rate equations with the nucleation-and-growth (NG) model (Eq (2)) at a finite time of 1 day. The lines show: the equilibrium critical assembly concentration ($\rho_{CAC}$, Eq (16), white '◇' symbols), the predicted threshold parameter values below which the median assembly timescale $\tau_{1/2}$ exceeds 1 day ($\rho_{nuc}$, Eq (29), red

'x' symbols), and the predicted locus of points corresponding to the minimum assembly timescale, beyond which monomer starvation begins to set in ($\rho_*$, Eq (32), black 'o' symbols). **(B)** The mass fraction of complete capsids $x_N$ as a function of total subunit concentration for no LLPS ($K_c = 1$). The line shows the equilibrium result (Eq (13)) and the symbols show results from numerically integrating the rate equations to 1 day ($\sim 9 \times 10^4$ sec, red 'x' symbols) and $t = 10^7$ seconds (blue o symbols). The dashed lines show $\rho_{CAC}$, $\rho_{nuc}$, and $\rho_*$. **(C)** Same as (B), but in the presence of LLPS, with $K_c = 36$. Other parameters in (A-C) are critical nucleus size $n_{nuc} = 3$, optimal size $N = 120$, subunit binding affinities $g_{nuc} = -4k_BT$, $g_{elong} = -17k_BT$, and compartment volume ratio $V_r = 10^{-3}$.

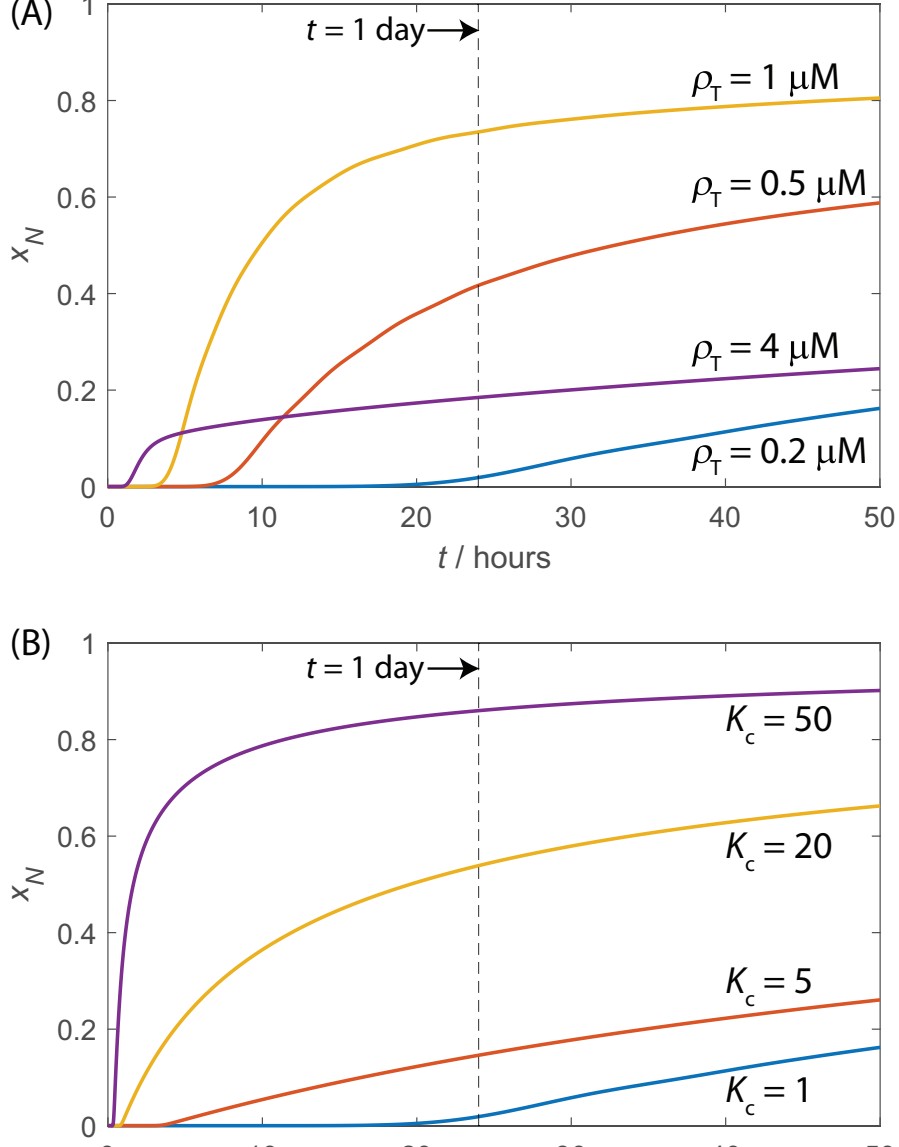

**Fig 4. The dependence of assembly kinetics on parameter values for the NG model with and without LLPS. (A)** The time evolution of the fraction of subunits in complete capsids $x_N$ for indicated values of the total subunit concentration $\rho_T$ computed from the Master equation, with no LLPS. **(B)** The time evolution of $x_N$ for indicated values of $K_c$, for fixed $\rho_T = 0.2\mu M$ and compartment ratio $V_r = 0.001$. The dashed lines show the timescale of 1 day corresponding to the red 'x' symbols in Fig 3B and 3C. Other parameter values for (A) and (B) are optimal assembly size $N = 120$, critical nucleus size $n_{nuc} = 3$, $g_{nuc} = -4k_BT$, $g_{elong} = -17k_BT$.

respectively show $x_N$ as a function of concentration measured at 1 day, $10^7$ seconds, and equilibrium. With or without LLPS, productive assembly at one day occurs over a much narrower range of concentrations than predicted by equilibrium, due to nucleation barriers at small concentrations and kinetic traps at high concentrations. Even at extremely long times the results have not reached full equilibrium due to kinetic traps at high concentrations. However, LLPS significantly broadens the range of concentrations leading to productive assembly at all timescales. The solid lines in Fig 3A and the dashed lines in Fig 3B and 3C indicate the CAC ($\rho_{CAC}$, Eq (16)), and scaling estimates for threshold concentrations below/above which productive assembly is impeded by large nucleation barriers or kinetic traps respectively (see section Assembly timescales without LLPS). Notice that in addition to making assembly more robust, LLPS also increases the maximum yield achievable at finite times. This increase arises because both nucleation and elongation rates can locally increase within the droplet due to the high local concentration while avoiding free subunit depletion.

The ability of LLPS to avoid kinetic traps arises because, for $V_r \ll 1$, the background acts like a buffer that steadily supplies free subunit to the compartment even when the nucleation rate is large. As a measure of this behavior, Fig 5 shows the concentration of subunits in the background normalized by the total concentration, $\rho_1^{bg}/\rho_T$ as a function of the maximum capsid formation rate (which occurs shortly after the end of the lag phase, before significant free subunit depletion has occurred). Here we have measured $\rho_1^{bg}$ at the time point corresponding to the maximum rate. Results are shown for LLPS assembly for the same parameters as in Fig 4, with increasing rates corresponding to increasing values of $K_c$. For the case without LLPS, we achieve faster rates by increasing the subunit-subunit affinities from ($g_{nuc}$, $g_{elong}$) = (−4,

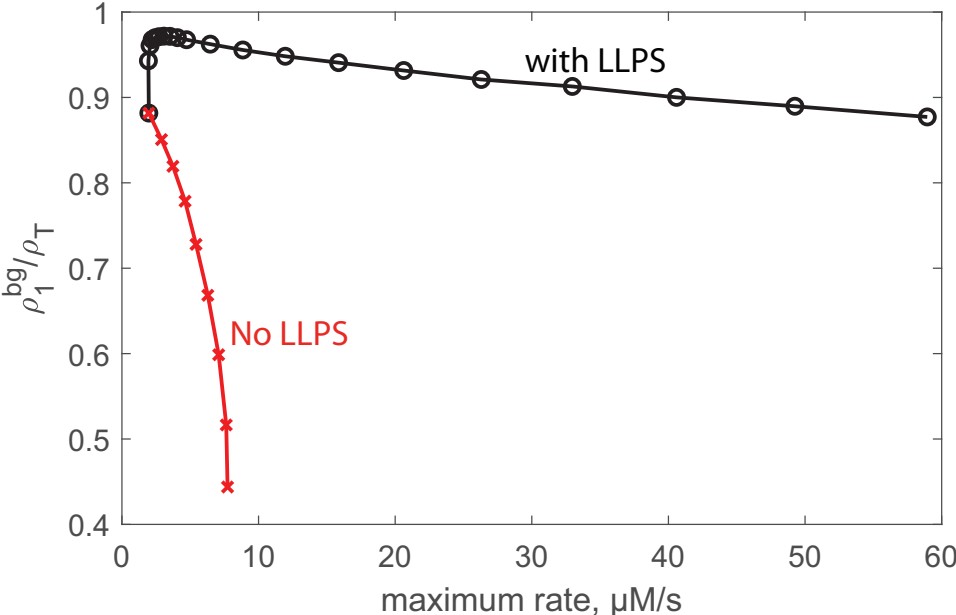

**Fig 5. The background acts as a buffer of free subunits for LLPS-dominated assembly.** The plot shows the concentration of subunits in the background, $\rho_1^{bg}$, as a function of the maximum capsid formation rate (maximized over time at a given set of parameter values) for assembly with LLPS (black 'o' symbols) and without LLPS (red 'x' symbols). For the LLPS case, the parameters correspond to those in Fig 4 with the increasing rate corresponding to $K_c$ ∈ [1, 30]. For the case without LLPS, the parameters are the same except that $K_c$ = 1 and the increasing rate is achieved by scaling the subunit-subunit affinities according to $s g_{nuc}$ and $s g_{elong}$ with $s$ ∈ [1, 1.5]. The results stop at $s$ = 1.5 because stronger affinities lead to kinetic traps and thus poor yields and a decreasing maximum rate.

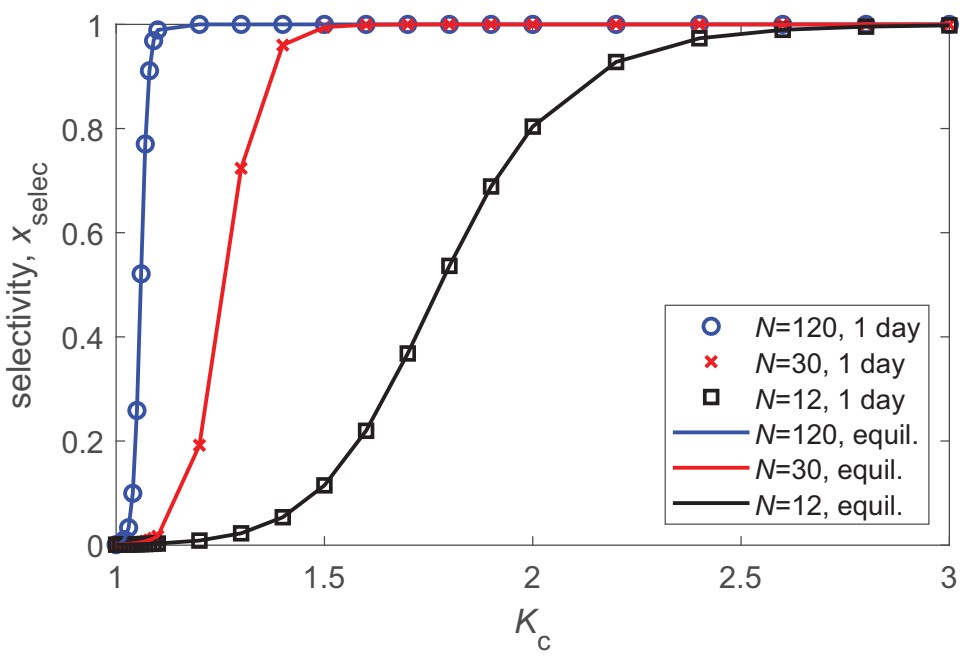

**Fig 6. Selectivity as a function of compartment partition coefficient and capsid size.** The symbols show the finite-time selectivity, $x_{\text{selec}} \equiv \frac{V_c \rho_N^c}{V_c \rho_N^c + V_{\text{bg}} \rho_N^{\text{bg}}}$, computed from the Master equation as a function of $K_c$ for three indicated values of the optimal assembly size $N$ at one day. The lines show the equilibrium selectivity (Eq (19)). Other parameters are the compartment volume ratio $V_r = 0.001$, $g_{\text{nuc}} = -4k_BT$, $g_{\text{elong}} = -17k_BT$, and $\rho_T = 0.2\mu$M.

$-17)k_BT$ to $(g_{\text{nuc}}, g_{\text{elong}}) = (-6, -25.5)k_BT$. We have increased affinities rather than total concentration (as we do for other results) to simplify comparison of $\rho_1^{\text{bg}}$ between the two cases. The results stop at $g_{\text{nuc}} = -6k_BT$ because stronger affinities lead to low yields and decreasing rates due to the monomer starvation trap.

We see that with LLPS the subunit concentration remains near $\rho_T$ even for extremely high assembly rates, whereas subunits are rapidly depleted without LLPS. For higher subunit affinities $|g_{\text{nuc}}| > 6k_BT$ depletion is so rapid that the monomer starvation trap sets in. Note that, in our Master equation description, subunits will eventually be depleted as $x_N \to 1$ even with LLPS, but in reality excluded volume effects (which are neglected in our model) would suppress assembly rates before this point unless complete capsids are expelled from the compartment.

**Selectivity.** Fig 6 shows the selectivity ($x_{\text{selec}}$) measured from Master equation solutions and the equilibrium result (Eq (19)) as a function of $K_c$ for several values of target capsid size $N$. We see that finite-time selectivity values closely match the equilibrium results, and that even an extremely small partition coefficient $K_c \gtrsim 2$ is sufficient to drive highly selective assembly in the compartment for large $N$.

## Scaling estimates for the effect Of LLPS on assembly timescales

To gain insights into how LLPS can affect assembly, in this section we derive simple scaling estimates for the timescales associated with the nucleation and growth mechanism of Eq (4). We closely follow Refs. [138, 149], but we extend the analysis to include the effect of a compartment. Although we introduce a number of simplifications, in the next section we show that the resulting scaling estimates provide good approximations when these simplifications are relaxed by numerically solving the Master equation models.

**Assembly timescales without LLPS.** Let us begin by summarizing the analysis of Ref [138] for assembly timescales in the absence of LLPS. As above, we consider a system of subunits with total concentration $\rho_T$ that form assemblies with optimal size $N$ subunits, and we break the assembly process into 'nucleation' and 'elongation' phases. For simplicity we assume that the association rate constant $f$ is independent of intermediate size (except where mentioned otherwise), so that the rates of association to each intermediate are $f\rho_1$.

We now write the time required for an individual assemblage to form as $\tau = \tau_{\text{nuc}} + \tau_{\text{elong}}$ with $\tau_{\text{nuc}}$ and $\tau_{\text{elong}}$ the average times for nucleation and elongation, respectively.

The *elongation* timescale can generally be estimated as [138, 146]

$$\tau_{\text{elong}} \cong N^\alpha / f\rho_1 \tag{20}$$

where we have assumed $N \gg n_{\text{nuc}}$ so that $N - n_{\text{nuc}} \approx N$. The factor in the numerator indicates that the elongation timescale increases with optimal assembly size (i.e. $\alpha > 0$) since $\mathcal{O}(N)$ independent subunit additions must occur. The value of the exponent $\alpha$ will depend on factors such as the dimensionality, the aggregate geometry, and the relative stability of intermediates, but we expect $1/2 \leq \alpha \leq 2$. For strongly forward-biased assembly during the elongation phase, $\alpha = 1$ for the NG model and $\alpha = 1/2$ for the CNT model (see [138] and Section B in S1 Text). Except where specified otherwise, for the scaling estimates in the rest of this article we set $\alpha = 1$, but the results are easily extendable to other exponent values.

The mean *nucleation* time at the beginning of the reaction can be estimated from the statistics of a random walk biased toward disassembly [29, 138], and can be approximately written as

$$\tau_{\text{nuc}} \approx f^{-1} \exp(G_{\hat{n}} / k_B T) \rho_T^{-n_{\text{nuc}}} \tag{21}$$

where $\hat{n} = n_{\text{nuc}} - 1$ so that $G_{\hat{n}}$ is the interaction free energy of the structure just below the critical nucleus size. The form of Eq (21) can be understood by noting that the pre-critical nucleus is present with concentration $\rho_{\hat{n}} \cong \exp(-G_{\hat{n}})\rho_T^{\hat{n}}$, and subunits associate to the precritical nucleus with rate $f\rho_T$. Note that the special case of $n_{\text{nuc}} = 2$ corresponds to no nucleation barrier (since two subunits must associate to begin assembly), in which case $G_{\hat{n}} = 0$ and $\tau_{\text{nuc}}(n_{\text{nuc}} = 2) \approx 1/f\rho_T^2$. We consider this case in Section D in S1 Text.

While Eq 21 gives the initial nucleation rate, the nucleation rate decreases over time due to subunit depletion, and asymptotically approaches zero as the concentration of complete capsids approaches its equilibrium value. Thus, we estimate the median assembly time $\tau_{1/2}$ (the time at which the reaction is 50% complete) by treating the system as a two-state reaction with $n_{\text{nuc}}$-th order kinetics, which yields [138]

$$\tau_{1/2} \cong \frac{A_{1/2} x_N}{Nf} \exp(G_{\hat{n}} / k_B T) \rho_T^{-\hat{n}} \tag{22}$$

with $A_{1/2} = \frac{2\hat{n}-1}{\hat{n}}$, and $x_N$ as the equilibrium fraction of subunits in complete capsids. The factor of $N^{-1}$ in Eq 22 accounts for the fact that $N$ subunits are depleted by each assembled capsid.

Analogous to crystallization or phase separation, there is a range of subunit concentrations and interaction strengths for which the unassembled state is metastable; i.e., the system is beyond the CAC so assembly is thermodynamically favorable, but the nucleation timescale exceeds experimentally relevant timescales. The boundary of this regime can be estimated by inverting Eq (22). Denoting the 'relevant' observation timescale as $\tau_{\text{obs}}$, we can estimate the

threshold subunit concentration below which nucleation will not be observed as

$$\rho_{\text{nuc}}^0 \cong \left(\frac{A_{1/2} x_N}{N f \tau_{\text{obs}}}\right)^{1/\hat{n}} \exp\left(\frac{G_{\hat{n}}}{\hat{n} k_{\text{B}} T}\right). \tag{23}$$

When elongation is fast compared to nucleation, the expressions Eq. S5 (in Section B in S1 Text) and Eq 22 respectively predict the duration of the lag phase and the median assembly time. However, these relations begin to fail above threshold values of the subunit concentration or subunit-subunit binding affinity, when nucleation and elongation timescales become comparable. Upon further increasing these parameters, nucleation becomes sufficiently fast that a significant fraction of monomers are depleted before elongation of most structures can complete. Subsequent evolution into complete assemblages then requires exchange of subunits between different intermediates (Ostwald ripening), which is an activated process and thus slow compared to assembly timescales. We describe this condition as the *monomer-starvation* kinetic trap. The threshold subunit concentration $\rho_*$ and interaction energies beyond which the system begins to enter the trap can be estimated by the locus of parameter values at which the median assembly time and elongation time are equal, i.e., $\tau_{1/2}(\rho_*) = \tau_{\text{elong}}(\rho_*)$:

$$\rho_*^0 \cong \left(\frac{A_{1/2} x_N}{N^2}\right)^{\frac{1}{\hat{n}-1}} \exp\left[\frac{G_{\hat{n}}}{(\hat{n}-1) k_{\text{B}} T}\right] \tag{24}$$

and a corresponding assembly timescale

$$\begin{aligned} \tau_{\text{min}}^0 &\equiv \tau_{\text{elong}}^0(\rho_*^0) \\ &\cong \frac{1}{f}\left(A_{1/2} x_N\right)^{\frac{-1}{\hat{n}-1}} \exp\left[\frac{-G_{\hat{n}}}{(\hat{n}-1) k_{\text{B}} T}\right] N^{\frac{\hat{n}+1}{\hat{n}-1}} \end{aligned} \tag{25}$$

Note that $\tau_{\text{min}}^0$ corresponds to approximately the minimal timescale or maximal assembly rate (over all subunit concentrations) since both $\tau_{1/2}$ and $\tau_{\text{elong}}$ monotonically decrease with subunit concentration before the onset of kinetic trapping.

**Assembly timescales with LLPS.** We now extend the scaling analysis to account for the presence of a compartment. Based on the conclusion of Section A in S1 Text that exchange of subunits between the background and compartment is typically much faster than assembly rates, we will make a quasi-equilibrium approximation for the relationship between subunit concentrations in the compartment and backround: $\rho_1^{\text{c}} = K_{\text{c}} K_{\text{eff}} \rho_{\text{T}}$ and $\rho_1^{\text{bg}} = K_{\text{eff}} \rho_{\text{T}}$.

As shown previously for irreversible aggregation [120], the compartment can dramatically amplify the nucleation rate by locally concentrating subunits. The total initial nucleation rate (in both the compartment and background at the beginning of the assembly process) is given by

$$\begin{aligned} r_{\text{nuc}}(V_{\text{r}}, K_{\text{c}}) &= s_{\text{nuc}} r_{\text{nuc}}^0 \\ s_{\text{nuc}} &\equiv K_{\text{eff}}^{n_{\text{nuc}}} \frac{1 + V_{\text{r}} K_{\text{c}}^{n_{\text{nuc}}}}{1 + V_{\text{r}}} \end{aligned} \tag{26}$$

with $r_{\text{nuc}}^0$ the nucleation rate in the absence of a compartment, and $s_{\text{nuc}}$ the acceleration factor for the initial nucleation rate. Eq (26) shows that for $K_{\text{c}}^{n_{\text{nuc}}} \gg 1/V_{\text{r}}$ nucleation will occur exclusively in the compartment, and the nucleation acceleration factor simplifies to $s_{\text{nuc}} \approx V_{\text{r}}/(V_{\text{r}} + 1/K_{\text{c}})^{n_{\text{nuc}}}$.

To estimate the parameters that maximize the initial nucleation rate, we optimize Eq (26) with respect to $V_r$ to obtain $V_r^* \approx \frac{1}{K_c \hat{n}}$ and thus a maximum nucleation acceleration of:

$$s_{\text{nuc}}^* \approx \frac{K_c^{\hat{n}}}{\hat{n} + 1}. \tag{27}$$

As in the equilibrium analysis in section Effects of LLPS on self-assembly equilibrium, under optimal conditions nucleation proceeds nearly as if the total subunit density were amplified by the partition coefficient $K_c$.

## Effect of LLPS on maximal assembly rates and kinetic traps

We now evaluate the effect of the compartment on the propensity of the system to undergo the monomer-starvation kinetic trap, by evaluating the dependence of the elongation and median assembly timescales on the phase-separation parameters.

The median assembly timescale can be computed by the same analysis used above for the nucleation time, leading to

$$\tau_{1/2}(V_r, K_c) = \tau_{1/2}^0 / s_{\text{nuc}}(V_r, K_c) \tag{28}$$

and similarly the threshold concentration below which nucleation does not occur on relevant timescales is

$$\rho_{\text{nuc}}(V_r, K_c) = \rho_{\text{nuc}}^0 / \left( s_{\text{nuc}}(V_r, K_c) \right)^{1/\hat{n}} \tag{29}$$

with $\rho_{\text{nuc}}^0$ given by Eq (23).

The elongation time within the compartment, $\tau_{\text{elong,D}}$ or background $\tau_{\text{elong,B}}$ is given by Eq. S5 in Section B in S1 Text with the appropriate local concentration $\rho_1^c = K_c K_{\text{eff}} \rho_T$ or $\rho_1^{\text{bg}} = K_{\text{eff}} \rho_T$. To estimate the onset of the kinetic trap, we must account for the numbers of capsids that are forming by both reaction channels (in the compartment or background), so we compute an average elongation time weighted by the relative number of assemblies that form the compartment or background

$$\tau_{\text{elong}} = \left[ \frac{\tau_{\text{elong,c}}^{-1} V_r K_c^N + \tau_{\text{elong,B}}^{-1}}{V_r K_c^N + 1} \right]^{-1} \tag{30}$$

In the limit of strongly forward-biased elongation and $K_c^{n_{\text{nuc}}} \gg 1/V_r$ so all nucleation occurs in the compartment, the elongation timescale is approximately

$$\tau_{\text{elong}}(K_c, V_r) \approx N / (f K_c K_{\text{eff}} \rho_T). \tag{31}$$

As discussed in section Assembly timescales without LLPS, the minimum assembly timescale occurs when the nucleation and elongation timescales are equal, $\tau_{\text{elong}}(K_c, V_r, \rho_*) = \tau_{1/2}(K_c, V_r, \rho_*)$; the monomer-starvation kinetic trap begins beyond this point. Using Eqs (26), (28), and (31) results in

$$\rho_*(K_c, V_r, n_{\text{nuc}}) = \rho_*^0 \left( \frac{K_c K_{\text{eff}}}{s_{\text{nuc}}} \right)^{\frac{1}{n_{\text{nuc}} - 2}}. \tag{32}$$

Finally, we can approximately extend the scaling estimates of this section to the CNT model by substituting Eq (7) for $n_{\text{nuc}}$.

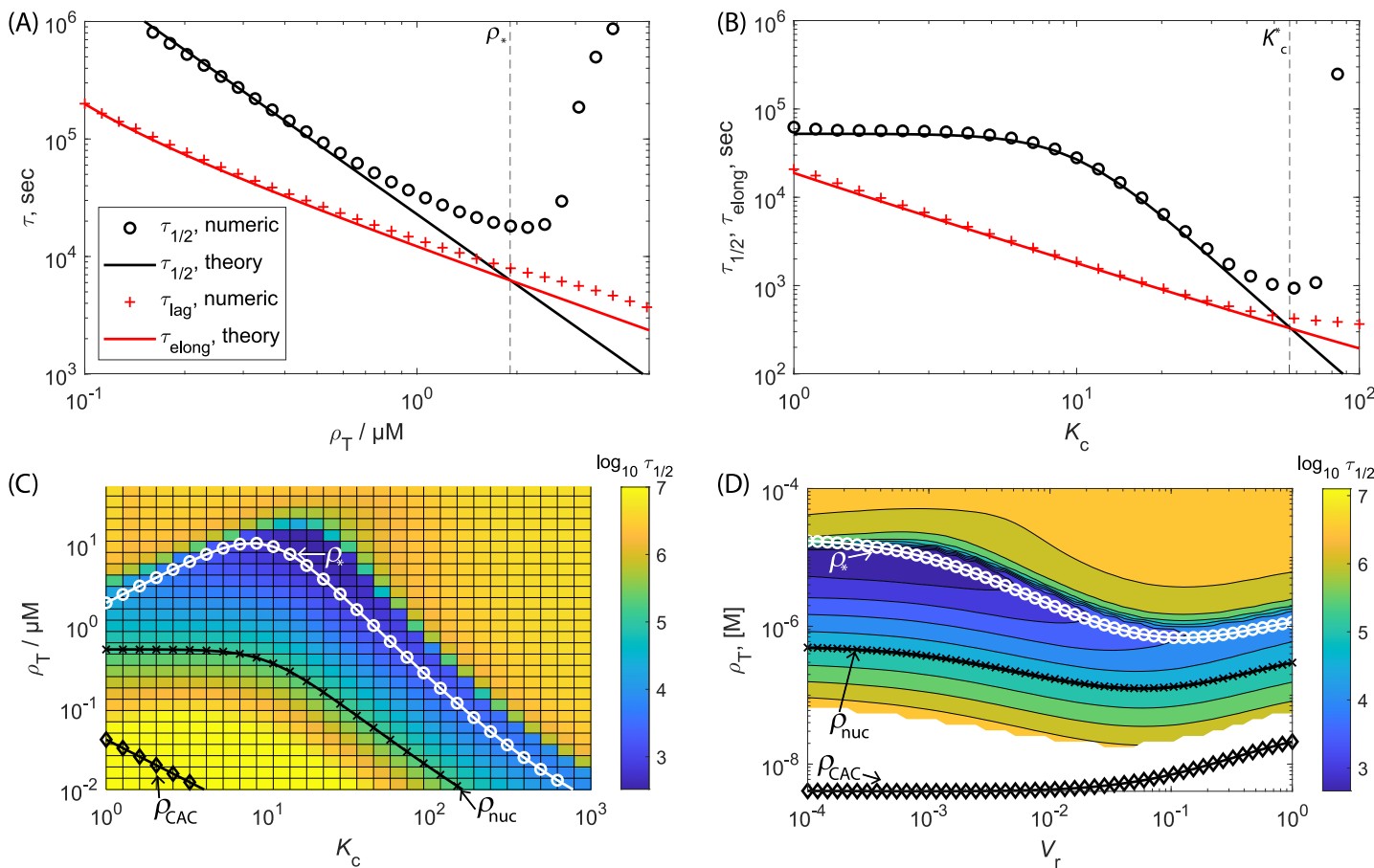

**Fig 7. Effect of LLPS on assembly timescales and the monomer-starvation kinetic trap for the NG model. (A)** The median assembly time $\tau_{1/2}$ and lag time calculated numerically from the Master equation (Eq (2)) and scaling estimates for the median assembly time (Eq (22)) and elongation time (Eq. S5 in Section B in S1 Text) as a function of subunit concentration, with no LLPS. The vertical dashed line indicates the scaling theory prediction for the concentration corresponding to the onset of the monomer starvation kinetic trap ($\rho_*$, Eq (32)). **(B)** Same quantities, shown as a function of the partition coefficient for concentration $\rho_T = 0.7 \mu M$. The vertical dashed line shows the estimate of the optimal value for the partition coefficient, $K_c^*$ (Eq. S8 in Section C in S1 Text). The compartment ratio is $V_r = 10^{-3}$ for (A) and (B). **(C, D)** The median assembly time predicted by the rate equation model as a function of subunit concentration and **(C)** varying compartment partition coefficient with $V_r = 10^{-3}$ or **(D)** varying $V_r$ with $K_c = 10$. The white line and 'o' symbols correspond to the theoretical prediction for the relationship between the subunit concentration and partition coefficient corresponding to the minimal assembly timescale (Eq (32)), beyond which the monomer-starvation kinetic trap begins to set in. The black line and 'x' symbols correspond to the relationship between the subunit concentration and $K_c$ value (Eq (29)) below which nucleation will not be observed on an experimentally relevant observation timescale of $\tau_{obs} = 1$ day. The black line and '◇' symbols denote the concentration and $K_c$ values corresponding to the CAC (Eq (16)). Other parameters are $N = 120$, $n_{nuc} = 3$, $g_{elong} = -17k_BT$, and $g_{nuc} = -4k_BT$.

Eq (32) shows that a key feature of preferential partitioning into the compartment is the ability of the system to buffer itself against the monomer-starvation kinetic trap while maintaining fast *localized* assembly in the compartment, as shown in Fig 5. We can further assess this feature in several ways as follows.

Fig 7A and 7B compare Eqs (28) and (30) to the median assembly and elongation times computed from the rate equations as a function of $\rho_T$ and $K_c$ respectively. We see that the scaling estimates and numerical results closely agree until the nucleation and elongation timescales become comparable; the threshold concentration $\rho_*$ (Eq (32)) and partition coefficient $K_c^*$ (Eq. S8 in Section C in S1 Text) are shown as vertical dashed lines in Fig 7A and 7B respectively. Fig 7C and 7D show the median assembly times computed from the rate equations as a function of $\rho_T$ and $K_c$ and $V_r$ respectively, with the locus of parameter values leading to minimum assembly predicted by Eq (32) shown as white o symbols. Fig 8 shows analogous results

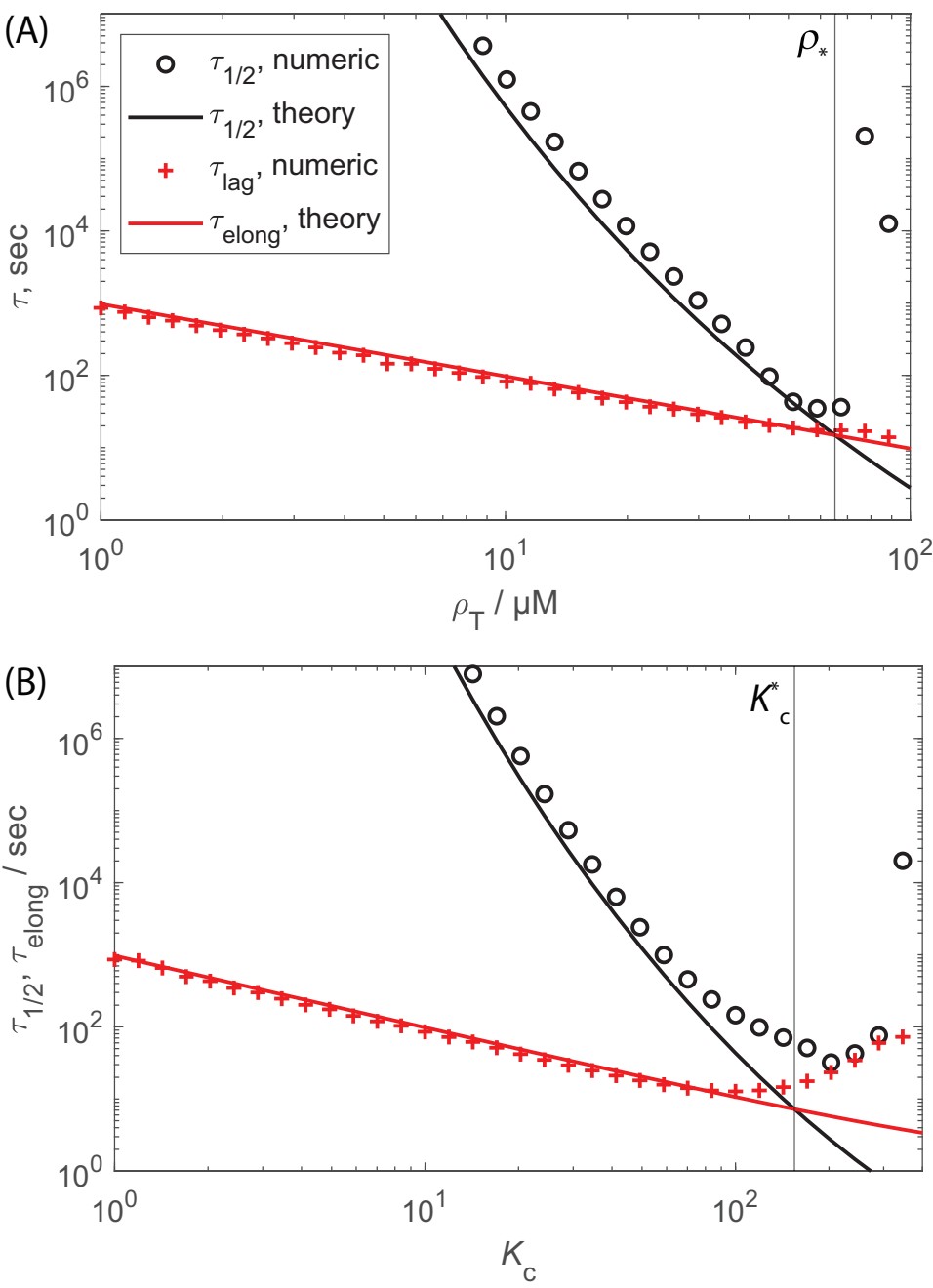

**Fig 8. Effect of LLPS on assembly timescales and the monomer-starvation kinetic trap for the CNT model. (A)** The median assembly time $\tau_{1/2}$ and lag time calculated numerically from the Master equation (Eq (2)) and scaling estimates for the median assembly time (Eq (22)) and elongation time (Eq. S5 in Section B in S1 Text) as a function of subunit concentration, with no LLPS. The vertical dashed line indicates the concentration corresponding to the onset of the monomer starvation kinetic trap ($\rho_*$, Eq (32)). **(B)** Same quantities, shown as a function of the partition coefficient for concentration $\rho_T = 0.6 \mu M$. The vertical dashed line shows the estimate of the optimal value for the partition coefficient, $K_c^*$ (Eq. S8 in Section C in S1 Text). Parameter values are $N = 120$, $g_{sub} = -17 k_B T$, and $V_r = 10^{-3}$.

for the CNT model. The prediction closely tracks the minimum assembly timescale observed in the numerical results. Below this threshold the median assembly time is closely predicted by Eq (28), with the assembly timescale sped up according to $s_{\mathrm{nuc}}$ in Eq (26). Above this threshold the numerically computed assembly timescales rapidly increase due to overly fast nucleation and thus onset of the monomer-starvation trap. We also show the CAC and the threshold for achieving assembly within an observation time of 1 day on these plots. Notice that, at a given value of $V_{\mathrm{r}}$, there is an optimal value of $K_{\mathrm{c}}$ (estimated below) which maximizes robustness of assembly to variations in concentration. In contrast, robustness monotonically increases with decreasing $V_{\mathrm{r}}$.

**The maximum assembly speedup depends on volume ratio, critical nucleus size, and subunit concentration.**  Given that the compartment both shifts and broadens the range of parameter values over which productive assembly can occur, it is of interest to determine parameters for which LLPS has the strongest effect on assembly times. To this end, we define the assembly 'speedup' as the factor by which the median assembly time decreases with LLPS relative to bulk solution: $s_{\mathrm{LLPS}}(K_{\mathrm{c}}, V_{\mathrm{r}}) \equiv \tau^0/\tau(K_{\mathrm{c}}, V_{\mathrm{r}})$. Recalling that the minimum assembly timescale occurs at $\rho_*$ when elongation and nucleation times are equal, we can then maximize the speedup with respect to the partition coefficient to obtain (Section C in S1 Text)

$$s_{\mathrm{LLPS}}^*(V_{\mathrm{r}}) \equiv \max_{K_{\mathrm{c}}} s_{\mathrm{LLPS}}(K_{\mathrm{c}}, V_{\mathrm{r}})$$

$$\approx V_{\mathrm{r}}^{-1/\hat{n}}\left(\frac{\rho_*^0}{\rho_{\mathrm{T}}}\right)^{\frac{\hat{n}^2-1}{\hat{n}}}. \tag{33}$$

Thus, for an optimal compartment partition coefficient, assembly can be sped up (i.e. $\tau_{1/2}$ reduced) by orders of magnitude for small $V_{\mathrm{r}}$. The degree of speedup increases with: decreasing $V_{\mathrm{r}}$, increasing critical nucleus size, and decreasing total subunit concentration. These trends can be understood as follows. Decreasing $V_{\mathrm{r}}$ means that subunits are not depleted as quickly within the compartment, thus allowing larger values of $K_{\mathrm{c}}$ and correspondingly higher local concentrations of subunits within the compartment without depleting subunits quickly enough to cause over-nucleation and monomer starvation. A larger critical nucleus size provides a larger separation between nucleation and growth timescales, thus enabling further concentration of subunits in the compartment without over-nucleation. The decreasing dependence on concentration arises because as the system approaches $\rho_*^0$, the assembly timescale without LLPS decreases and thus so does the extent of possible speedup before over-nucleation sets in. However, note in Fig 8 that the maximum optimal concentration in the presence of LLPS exceeds the intrinsic value, $\rho_* > \rho_*^0$, due to the extra regulation of nucleation and growth timescales allowed by a compartment. Also note that LLPS provides speedup even after monomer starvation begins to set in.

Fig 9 compares the scaling estimate for speedup (see Eq. S9 in Section C in S1 Text) to the value computed numerically from the rate equations. For the numerical value, we computed $\tau_{1/2}^0$ for fixed $\rho_{\mathrm{T}}$ and interaction parameters by numerically integrating the rate equations without LLPS, and then performed numerical minimization over $K_{\mathrm{c}}$ to obtain $\tau_{1/2}^*(V_{\mathrm{r}}) = \min_{K_{\mathrm{c}}} \tau_{1/2}(V_{\mathrm{r}}, K_{\mathrm{c}})$ with respect to $K_{\mathrm{c}}$ for the same $\rho_{\mathrm{T}}$ and interaction parameters. Then the speedup is given by $s_{\mathrm{LLPS}}^*(V_{\mathrm{r}}, \rho_{\mathrm{T}}) = \tau_{1/2}^0(\rho_{\mathrm{T}})/\tau_{1/2}^*(V_{\mathrm{r}}, \rho_{\mathrm{T}})$. We have presented the speedup as a function of concentration normalized by the optimal value in the absence of LLPS so that the results can be shown on the same plot. As shown in Fig 9A, the scaling estimate closely matches the numerical result until $\rho_{\mathrm{T}}$ exceeds the maximum value of $\rho_*$ at which point monomer starvation begins to set in. The results for the CNT model are obtained by substituting Eq (7) into Eqs. S7-S9. The agreement is reasonable but not as close as the NG estimate because

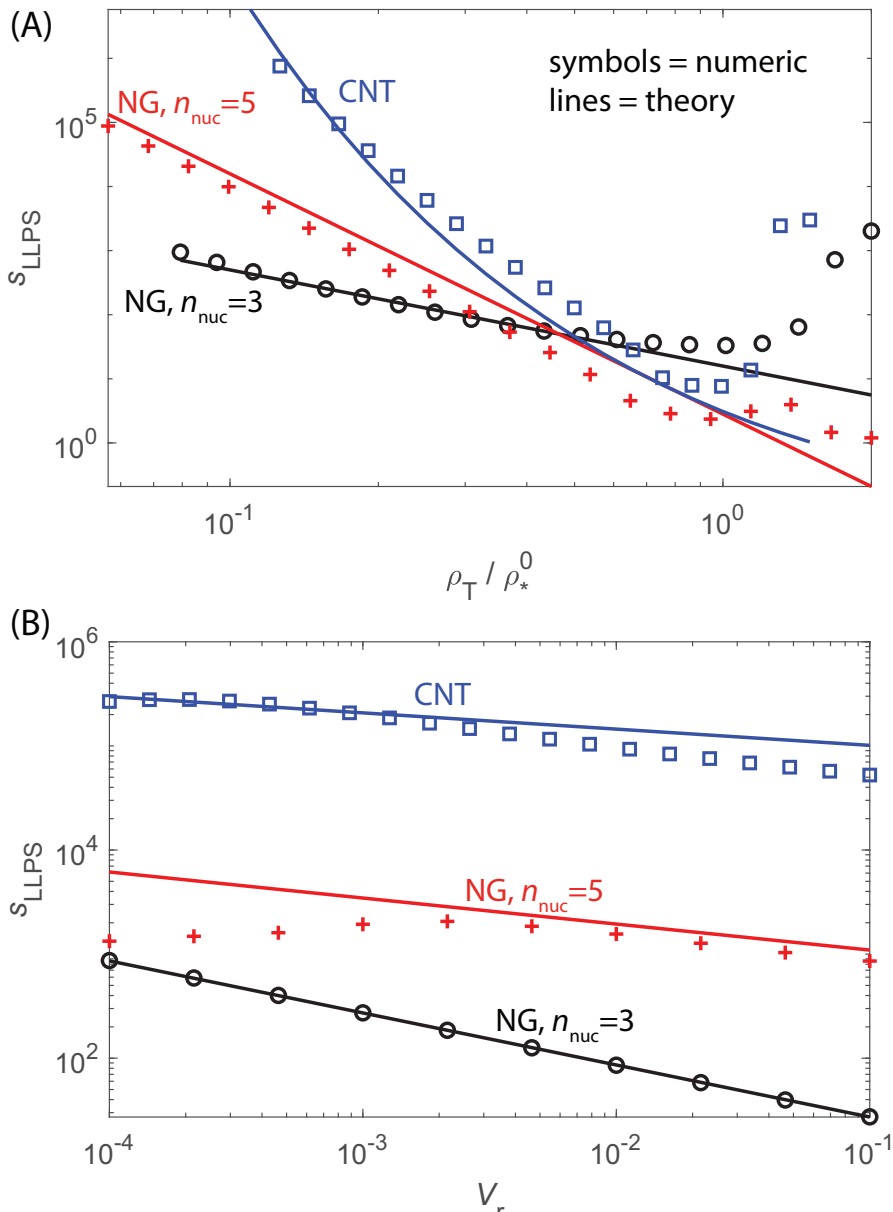

**Fig 9. Maximum speedup provided by LLPS accounting for kinetic trapping.** (A) The assembly speedup optimized over the compartment partition coefficient, $s_{LLPS}^*(V_r) = \min_{K_c} \tau_{min}^0 / \tau_{min}(K_c, V_r)$, is shown as a function of subunit concentration $\rho_T$ for fixed $V_r = 10^{-3}$. Results are shown for the NG model with critical nucleus sizes $n_{nuc} = 3$ and $n_{nuc} = 5$, as well as the CNT model. The symbols show results obtained from the Master equation with $\tau_{min}$ calculated by numerically minimizing $\tau_{1/2}$ with respect to $K_c$. The lines show the approximate estimate Eq. S9 in Section C in S1 Text. The subunit concentrations are scaled by the optimal concentration in the absence of LLPS, $\rho_*^0$, so that the results are visible on a single plot. The optimal concentrations for these parameters are $\rho_*^0(n_{nuc} = 3) = 1.9\,\mu M$, $\rho_*^0(n_{nuc} = 5) = 1.2$ mM for the NG model and $\rho_*^0 = 65\,\mu M$ for the CNT model. (B) The assembly speedup optimized over $K_c$ as a function of $V_r$ for fixed subunit concentration $\rho_T/\rho_* = 0.15$.

Eq (7) is based on the critical nucleus size in the absence of LLPS. As noted above, we see that LLPS continues to speedup the assembly time even in the monomer-starvation regime.

Fig 9B shows the speedup as a function of $V_r$ for fixed $\rho_T$. Here we see good agreement between the scaling estimate and numerical results, except the numerical speedup diminishes at small $V_r$ for $n_{nuc} = 5$. This occurs because the minimum assembly timescale has decreased below the diffusion limited timescale (Eq. S4 in Section A in S1 Text) in this regime, which is not accounted for in the scaling estimate.

**Maximizing assembly robustness.** Notice that $K_c^*$ decreases with increasing subunit concentration (see Eq. S7 in Section C in S1 Text). For large subunit concentrations (i.e. $\rho_T \to \rho_*^0$), the assumption that assembly occurs primarily in the compartment breaks down and we must consider the full form of $s_{nuc}$ (Eq (26)). If we substitute this into the expression for Eq (32), we see that there is a maximum in $\rho_*$ at

$$K_c^{**} \approx \left(\frac{1}{V_r \hat{n}}\right)^{1/(\hat{n}+1)} \tag{34}$$

which results in

$$\frac{\rho_*(K_c^{**}, V_r)}{\rho_*^0} \sim V_r^{-1/(\hat{n}^2-1)}. \tag{35}$$

Finally, using Eq 16 shows that the range of subunit concentrations over which assembly is favorable increases as

$$\frac{\rho_*/\rho_{CAC}}{\rho_*^0/\rho_{CAC}^0} \sim V_r^{-n/(\hat{n}^2-1)}. \tag{36}$$

For small $V_r$, the width of this range, and thus the robustness of assembly to variations in subunit concentration or subunit-subunit binding affinities, increases by orders of magnitude (Fig 7D).

We can alternatively specify robustness by defining the region of *productive assembly* as the set of parameter values for which nucleation occurs within experimentally relevant timescales (e.g. 1 day) and avoids the monomer-starvation trap. To maximize the breadth of this range, we define $K_c^{nuc}$ as the partition coefficient that maximizes the ratio of the monomer-starvation threshold to the nucleation timescale threshold (Eq (29)): $K_c^{nuc} \approx \left(\frac{\hat{n}}{V_r}\right)^{1/(\hat{n}+1)}$.

## Conclusions

### Summary

It is well-established that efficient self-assembly in homogeneous solution is constrained to a narrow window of moderate subunit concentrations and interaction strengths, due to the competing constraints of minimizing nucleation timescales while avoiding kinetic traps [28–47]. Here, we find that when subunits preferentially partition into nano- or microscale compartments, the range of parameters leading to productive assembly can be broadened by more than an order of magnitude, and the corresponding assembly timescales can be reduced by multiple orders of magnitude. Moreover, in part of this parameter range, almost all assembly occurs within the compartment interior, thus allowing spatial control over assembly. These behaviors depend sensitively on two parameters that control phase coexistence: the partition coefficient of subunits into phase separated compartments and the size ratio between the compartments and the cell. In addition, we find that the maximum degree of speedup due to LLPS

increases with: decreasing compartment/cell size ratio or subunit concentration and increasing assembly critical nucleus size.

These effects arise because the compartment (or compartments) drive high local concentrations of subunits, thus minimizing the local nucleation timescales, but the small size of the compartment limits the total nucleation rate (averaged over the whole system volume). In effect, the bulk exterior acts as a subunit 'buffer' that, early in the reaction, steadily supplies subunits to the compartment and thereby suppresses the monomer starvation kinetic trap (see Fig 5). This mechanism has the strongest effect on robustness of assembly to variations in parameter values for small critical nucleus sizes or non-nucleated reactions, for which the homogeneous system lacks an intrinsic difference between nucleation and growth timescales and thus is most sensitive to subunit depletion. However, the decrease in assembly timescales is most dramatic for larger critical nucleus sizes, due to the high-order dependence of assembly timescales on local subunit concentration.

## Relevant parameter ranges

Since these mechanisms depend on localization of subunits, the ability of LLPS to control assembly increases with decreasing compartment size (relative to the total system size). To estimate the relevance of this effect in biological systems, consider that typical compartments in eukaryotic cells range in size from $\sim$50nm to 10 $\mu$m [54, 123, 124, 130]. For a compartment with diameter 1 $\mu$m in a cell with diameter 20 $\mu$m, the volume ratio of the compartment relative to bulk is $V_r \sim 10^{-4}$. From Eqs (35) and (36) and Fig 7, we see that the range of subunit concentrations leading to productive assembly could increase by up to two orders of magnitude, with increases in assembly rates exceeding five orders of magnitude (Eq. S9 in Section C in S1 Text and Fig 9). These increases reflect the ability of compartmentalization to enable fast localized assembly while minimizing the rate of global depletion of subunits.

## Testing in experiments

Since our models are general, the quantitative predictions and scaling formulae described here can apply to a broad range of experimental systems in which there is phase coexistence and the assembly subunits preferentially partition into one phase. Such phase-separated compartments appear to be ubiquitous in cells, and as noted in the introduction, assembly of diverse structures such as clathrin cages, actin filaments, and neuronal synapses can occur within compartments. The systems which most directly inspired this work are the phase-separated compartments generated during viral infections (e.g. virus factories, replication sites, Negri bodies, inclusion bodies, or viroplasms [88–108]), within which viral particles undergo assembly. However, directly testing our theoretical predictions may be easier in *in vitro* experiments, since there is a greater ability to control the size and composition of compartments [52, 54, 76, 99]. Compartment sizes can be controlled in bulk systems by varying the total density of the phase-separating components, while microfluidic arrays enable precise control over droplet sizes and compositions.

## Outlook

We have focused on a minimal model for this first study of the effects of LLPS on assembly robustness. There are a number of additional physical ingredients that merit further exploration. For parameter regimes that lead to high subunit concentrations within the compartment, the assumption that the subunits do not affect the equilibrium compartment size and composition will break down. Importantly, the rate equation models and scaling estimates considered here do not account for kinetic traps resulting from malformed assemblies, which can arise

when subunits bound with incorrect geometries do not have time to anneal before becoming locked into place by association of additional subunits (e.g. [11, 32, 39, 41, 42, 141, 142, 150, 151]). Since association rates increase with concentration, we anticipate that malformed assemblies will occur above a threshold local concentration within the compartment, thus limiting the maximum speed up provided by LLPS. This threshold concentration increases with the geometric specificity of the subunit-subunit interactions. Thus, for sufficiently specific interactions the results described here will not qualitatively change when accounting for malformed structures—there will be a significant range of local concentrations, and thus assembly speed up, before either the malformed structure for monomer starvation kinetic trap set in. We will explore the effects of malformed assemblies on LLPS-coupled assembly in a future work.

Further, Schmit and Michaels showed that, if subunit diffusion slows with increasing subunit-compartment attraction strength ($g_c$ and $K_c$ in our model), then there is an optimum $K_c$ beyond which assembly slows. The results in our work arising from competing interactions are distinct from this effect. Other important effects to be incorporated include: slow diffusion into/out of the compartment [137], accounting for spatial structure and stochasticity of assembly [32, 152–155], nonequilibrium effects such as synthesis of new subunits or phosphorylization-driven changes in assembly activity, selective partitioning of different species in a multicomponent assembly reaction, and the ability of the compartment to template the size and shape of assemblies, such as occurs in bacterial microcompartments [43, 45, 132, 133].

Ultimately, understanding how different combinations of these physical mechanisms enable phase separation processes to control the time, place, and rate of assembly will engender a more complete understanding of biological self-assembly, and can advance strategies for designing human-engineered nanostructured materials.

## Supporting information

**S1 Text.** Section A: The kinetics of subunits and assemblies partitioning between the compartment and background. Section B: Scaling estimates of elongation timescales. Section C: Scaling estimates of maximum assembly speedup. Section D: Analysis for $n_{\mathrm{nuc}} = 2$.
(PDF)

## Acknowledgments

Computational resources were provided by NSF XSEDE computing resources (Expanse) and the Brandeis HPCC which is by the NSF through DMR-MRSEC 2011846 and OAC-1920147. We gratefully acknowledge John Patton for useful discussions and William Jacobs for insightful comments on the manuscript.

## Author Contributions

**Conceptualization:** Michael F. Hagan.

**Data curation:** Michael F. Hagan, Farzaneh Mohajerani.

**Formal analysis:** Michael F. Hagan.

**Funding acquisition:** Michael F. Hagan, Farzaneh Mohajerani.

**Investigation:** Michael F. Hagan, Farzaneh Mohajerani.

**Methodology:** Michael F. Hagan, Farzaneh Mohajerani.

**Project administration:** Michael F. Hagan.

**Resources:** Michael F. Hagan.

**Software:** Michael F. Hagan, Farzaneh Mohajerani.

**Supervision:** Michael F. Hagan.

**Validation:** Michael F. Hagan, Farzaneh Mohajerani.

**Visualization:** Michael F. Hagan, Farzaneh Mohajerani.

**Writing – original draft:** Michael F. Hagan, Farzaneh Mohajerani.

**Writing – review & editing:** Michael F. Hagan, Farzaneh Mohajerani.

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
