## [Decision Letter · Decision Letter 0]

23 Dec 2022

Dear Dr. Hagan,

Thank you very much for submitting your manuscript "Self-Assembly Coupled to Liquid-Liquid Phase Separation" for consideration at PLOS Computational Biology.

As with all papers reviewed by the journal, your manuscript was reviewed by members of the editorial board and by several independent reviewers. In light of the reviews (below this email), we would like to invite the resubmission of a significantly-revised version that takes into account the reviewers' comments.

We cannot make any decision about publication until we have seen the revised manuscript and your response to the reviewers' comments. Your revised manuscript is also likely to be sent to reviewers for further evaluation.

Sincerely,

Huan-Xiang Zhou

Guest Editor

PLOS Computational Biology

Arne Elofsson

Section Editor

PLOS Computational Biology

Reviewer's Responses to Questions

**Comments to the Authors:**

Reviewer #1: Comments are attached.

Reviewer #2: This is an interesting theoretical development that addresses the thermodynamic and dynamic ramifications for the modulation of viral-capsid-like assembly in cellular compartments that maintain a higher concentration of subunits. The results are suitable for publication in PLoS CB, but revision is necessary to clarify/address the following issues:

1. The authors elect to use “domain” to describe the compartment that maintain a higher concentration of subunits. This term is not ideal if the authors aim to frame their investigation in the context of recent advances in the study of biomolecular condensates (which is clearly the case judging from the large number of references in this area that the authors included). These condensates are usually referred to as “compartments”. In this very field that the authors aim to contextualize their work, “domain” usually refers to protein domains. So the authors’ terminology will not help facilitate understanding by readers who are familiar with the biomolecular condensate terminology. I suggest that the authors to change “domain” to “compartment” (they did use “compartmentalization” on p.12 of the manuscript). Failing that, they should at least state clearly at the outset that their “domain” is equivalent to intracellular membraneless biomolecular compartments.

2. Though it is not stated explicitly in the text, the formulation in Eqs.13-14 indicates that subunits can only be added one-by-one in the authors’ model. In other words, associations of, say, two n=2 complexes is postulated to be impossible (because there are no such terms in Eqs.13-14). Is this the case? If so, this limitation has to be justified. Does it have an experimental basis in viral capsid assembly? Irrespective of whether viral capsid assembly obeys this rule, the authors should discuss (and/or show additional calculation) what will happen when this limitation is lifted.

3. Notation problem: fN is used to denote fraction of completed assembly in Fig.2 and Fig.3 but is also used (on the same page) to represent associate rate constant for the n=N state in Eq.14. This situation should be rectified. Authors should use different symbols for different quantities in the same paper, especially when different meanings of the same symbols are invoked so close to each other in the text.

4. Additional information should be provided in Fig.3 to show corresponding data for ρT > 1μM (by adding a figure or adding parts to this figure or simply by adding curves to the existing figure). This is important for illustrating the decreases seen for the red curves in Fig.2 for larger ρT , which is an interesting finding. To make this correspondence clearer, vertical dashed lines should be added to indicate t = 1 day time point (24 hr) along the horizontal scales of these plots.

5. It is important to emphasize that the kinetic traps mentioned in the present work are caused by depletion of subunits but not “malformed assembly” (mentioned in page 12). Malformed assemblies can also cause kinetic traps (as their disassembly requires an energetic cost). This matter should be commented upon with a clarification about the basic difference between these two types of kinetic traps.

6. More specific examples should be provided in the concluding discussion about what real biophysical process this formulation is expected to be applicable to. What kind of viral capsid? Which intracellular compartments (domains) are the likely locations for viral capsid assembly?

7. The authors clearly aim to be rather comprehensive in their citing literature in the Introduction. In this regard, references for several key advances in the theoretical study of biomolecular condensates are missing, which should be included in the revised version of this manuscript, as follows: Lin et al., Phys Rev Lett 117:178101 (2016) (first theory for sequence-dependent biomolecular phase separation ) should be included around ref.[50], and Lin & Chan, Biophys J 112:2043-2046 (2017); Amin et al., J Phys Chem B 124:6709-6720 (2020) – two theoretical works that address the relationship between single-chain and two-chain association properties and multiple-chain phase separation propensity – should be included around ref.[63].

8. Likely typographical errors:

(i) 3rd line below Eq.10: “Kdom → 0” should be “Kdom → 1”?

(ii) Eq.A3: subscript n is missing in the two diffusion coefficient. Judging from these typos, there are likely others – this referee did not do an exhaustive check. The authors are strongly encouraged to double-check to make sure that their notation is clear and consistent and without omissions and typos.

**Have the authors made all data and (if applicable) computational code underlying the findings in their manuscript fully available?**

Reviewer #1: Yes

Reviewer #2: Yes

PLOS authors have the option to publish the peer review history of their article (what does this mean?). If published, this will include your full peer review and any attached files.

Reviewer #1: No

Reviewer #2: No
---

## [Decision Letter · Decision Letter 1]

18 Apr 2023

Dear Dr. Hagan,

Thank you very much for submitting your manuscript "Self-Assembly Coupled to Liquid-Liquid Phase Separation" for consideration at PLOS Computational Biology. As with all papers reviewed by the journal, your manuscript was reviewed by members of the editorial board and by several independent reviewers. The reviewers appreciated the attention to an important topic. Based on the reviews, we are likely to accept this manuscript for publication, providing that you modify the manuscript according to the review recommendations.

Please just update the colors in Fig 1.

Sincerely,

Arne Elofsson

Section Editor

PLOS Computational Biology

Arne Elofsson

Section Editor

PLOS Computational Biology

Please just update the colors in Fig 1.

Reviewer's Responses to Questions

**Comments to the Authors:**

Reviewer #1: The authors have made a very good attempt at addressing the comments raised in the first review. Their explanations are more thorough and the changes have significantly improved the clarity and scope of the paper. One additional comment pertains to figure 1; there, the color scheme used for the text (i.e., on the compartment) makes for low readability. The authors should consider a different color palette as would make the figure more accessible.

Reviewer #2: The authors have adequately (in fact quite thoroughly) addressed my previous concerns. I recommend publication of the current revised version of this manuscript.

**Have the authors made all data and (if applicable) computational code underlying the findings in their manuscript fully available?**

Reviewer #1: Yes

Reviewer #2: Yes

PLOS authors have the option to publish the peer review history of their article (what does this mean?). If published, this will include your full peer review and any attached files.

Reviewer #1: No

Reviewer #2: **Yes: **Hue Sun Chan

Figure Files:

Data Requirements:

Reproducibility:

References:

---

## [Editor Report · Decision Letter 2]

26 Apr 2023

Dear Dr. Hagan,

We are pleased to inform you that your manuscript 'Self-Assembly Coupled to Liquid-Liquid Phase Separation' has been provisionally accepted for publication in PLOS Computational Biology.

Best regards,

Huan-Xiang Zhou

Guest Editor

PLOS Computational Biology

Arne Elofsson

Section Editor

PLOS Computational Biology

---

## [Editor Report · Acceptance letter]

9 May 2023

PCOMPBIOL-D-22-01510R2 

Self-Assembly Coupled to Liquid-Liquid Phase Separation

Dear Dr Hagan,

I am pleased to inform you that your manuscript has been formally accepted for publication in PLOS Computational Biology. Your manuscript is now with our production department and you will be notified of the publication date in due course.

With kind regards,

Zsofia Freund
